

# 1  The status and challenge of global fire modelling

S. Hantson[1], A. Arneth[1], S. P. Harrison[2,3], D. I. Kelley[2,3], I. C. Prentice[3,4], S. S. Rabin[5], S. Archibald[6,7], F.
Mouillot[8], S. R. Arnold[9], P. Artaxo[10], D. Bachelet[11,12], P. Ciais[13], M. Forrest[14], P. Friedlingstein[15], T.
Hickler[14,16], J. O. Kaplan[17], S. Kloster[18], W. Knorr[19], G. Lasslop[18], F. Li[20], S. Mangeon[21], J. R. Melton[22], A.
Meyn[23], S. Sitch[24], A. Spessa[25,26], G. R. van der Werf[27], A. Voulgarakis[21], C. Yue[13].
[1] Karlsruhe Institute of Technology, Institute of Meteorology and Climate research, Atmospheric Environmental
Research, 82467 Garmisch-Partenkirchen, Germany.
[2] School of Archaeology, Geography and Environmental Sciences (SAGES), University of Reading, Reading, UK.
[3] School of Biological Sciences, Macquarie University, North Ryde, NSW 2109, Australia.
[4] AXA Chair of Biosphere and Climate Impacts, Grand Challenges in Ecosystem and the Environment, Department
of Life Sciences and Grantham Institute – Climate Change and the Environment, Imperial College London, Silwood
Park Campus, Buckhurst Road, Ascot SL5 7PY, UK.
[5] Department of Ecology & Evolutionary Biology, Princeton University, Princeton, NJ, USA.
[6] School of Animal, Plant and Environmental Sciences, University of the Witwatersrand, Johannesburg 2050, South
Africa.
[7] Natural Resources and the Environment, CSIR, PO Box 395, Pretoria, 0001, South Africa.
[8] UMR 5175 CEFE, CNRS/Université de Montpellier/Université Paul-Valéry Montpellier/EPHE/IRD, 1919 route de
Mende, 34293 Montpellier Cedex 5, France.
[9] Institute for Climate and Atmospheric Science, School of Earth & Environment, University of Leeds, UK.
[10] Institute of Physics, University of São Paulo, Rua do Matão, Travessa R, 187, CEP05508-090, São Paulo, S.P.,
Brazil.
[11] Biological and Ecological Engineering, Oregon State University, Corvallis, OR 97331, USA.
[12] Conservation Biology Institute, 136 SW Washington Ave., Suite 202, Corvallis, OR 97333, USA.
[13] Laboratoire des Sciences du Climat et de l'Environnement, LSCE/IPSL, CEA-CNRS-UVSQ, Université Paris-
Saclay, F-91198 Gif-sur-Yvette, France.
[14] Senckenberg Biodiversity and Climate Research Institute (BiK-F), Senckenberganlage 25, 60325 Frankfurt am
Main, Germany.
[15] College of Engineering Mathematics and Physical Sciences, University of Exeter, Exeter, United Kingdom.
[16] Institute of Physical Geography, Goethe-University, Altenhöferallee 1, 60438 Frankfurt am Main, Germany.
[17] Institute of Earth Surface Dynamics, University of Lausanne, 1015 Lausanne, Switzerland.
[18] Max Planck Institute for Meteorology, Bundesstraße 53, 20164 Hamburg, Germany.
[19] Department of Physical Geography and Ecosystem Science, Lund University, 22362Lund, Sweden.
[20] International Center for Climate and Environmental Sciences, Institute of Atmospheric Physics, Chinese Academy
of Sciences, Beijing, China.
[21] Department of Physics, Imperial College London, London, United Kingdom.
[22] Climate Research Division, Environment Canada, Victoria, BC, V8W 2Y2, Canada.
[23] Karlsruhe Institute of Technology, Atmosphere and Climate Programme, 76344 Eggenstein-Leopoldshafen,
Germany .
[24] College of Life and Environmental Sciences, University of Exeter, Exeter EX4 4RJ, United Kingdom.
[25] Department of Environment, Earth and Ecosystems, Open University, Milton Keynes, United Kingdom.
[26] Department Atmospheric Chemistry, Max Planck Institute for Chemistry, Mainz, Germany.
[27] Faculty of Earth and Life Sciences, VU University Amsterdam, De Boelelaan 1085, 1081HV, Amsterdam, The
Netherlands.

*Correspondence to*: S. Hantson (stijn.hantson@kit.edu)



**Abstract.** Biomass burning impacts vegetation dynamics, biogeochemical cycling, atmospheric chemistry, and
climate, with sometimes deleterious socio-economic impacts. Under future climate projections it is often expected
that the risk of wildfires will increase. Our ability to predict the magnitude and geographic pattern of future fire
impacts rests on our ability to model fire regimes, either using well-founded empirical relationships or process-based
models with good predictive skill. A large variety of models exist today and it is still unclear which type of model or
degree of complexity is required to model fire adequately at regional to global scales. This is the central question
underpinning the creation of the Fire Model Intercomparison Project - FireMIP, an international project to compare
and evaluate existing global fire models against benchmark data sets for present-day and historical conditions. In this
paper we summarise the current state-of-the-art in fire regime modelling and model evaluation, and outline what
lessons may be learned from FireMIP.

## 1. Introduction
Each year, about 4% of the global vegetated area is burned (Giglio et al., 2013; Randerson et al., 2012). Fire is the
most important type of disturbance and as such is a key driver of vegetation dynamics (Bond et al., 2005), both in
terms of succession and in maintaining fire-adapted ecosystems (Furley et al., 2008; Staver et al., 2011; Hirota et al.,
2011; Rogers et al., 2015). Fires play an essential role in ecosystem functioning, species diversity, plant community
structure and carbon storage. The impact fire has on the ecosystem depends on the local fire regime, including fire
frequency, intensity, seasonality etc. Fire is also important through its effect on radiative forcing, biogeochemical
cycling and biogeophysical effects (Bond-Lamberty et al., 2007; Bowman et al., 2009; Ward et al., 2012, Yue et al.,
67 2015).

Global carbon dioxide emissions from biomass burning are estimated to be about 2 PgC (P = $10^{15}$) per year of which
approximately 0.6 PgC/yr comes from tropical deforestation and peat fires (van der Werf et al., 2010). This is
equivalent to ca 25% of those from fossil fuel combustion (Ciais et al., 2013; Boden et al., 2013), although a
significant fraction of these emissions is taken up during vegetation regrowth after fire. Together, fire significantly
decreases the net carbon gain of global terrestrial ecosystems by 1.0 Pg C $yr^{-1}$ averaged across the 20th century (Li et
al., 2014). Fire emissions are also an important driver of inter-annual variability in the atmospheric growth rate of
$CO_2$ (van der Werf et al., 2004; van der Werf et al., 2010; Prentice et al., 2011; Guerlet et al., 2013) and a significant
contribution to the atmospheric budgets of $CH_4$, CO and many other atmospheric constituents. As a source of aerosol
(including black carbon) and ozone precursors (Voulgarakis and Field, 2015), emissions from fires contribute
directly and indirectly to radiative forcing (Myhre et al., 2013; Ward et al., 2012), reducing net shortwave radiation
at the surface and warming the lower atmosphere, thus affecting regional temperature, clouds, and precipitation
(Tosca et al., 2010; Tosca et al., 2014; Ten Hoeve et al., 2012; Boucher et al., 2013) and regional to large-scale
atmospheric circulation patterns (Tosca et al., 2013; Zhang et al., 2009). Through their impacts on ozone, and as a
source of CO and other volatile organic compounds, fires also affect the atmospheric abundance of the OH radical,
which determines the atmospheric lifetime of the greenhouse gas methane (Bousquet et al., 2006). In addition, ozone





produced from fires is directly harmful to plants, reducing photosynthesis (Pacifico et al., 2015) and fire-emitted
aerosol can shift the balance between diffuse and direct radiation (Mercado et al., 2009; Cirino et al., 2014).
Deposition of fire produced N- (Chen et al., 2010) and P-aerosols (Wang et al., 2015) can enhance productivity in
nutrient limited ecosystems.
Fire also has direct effects on human society: more than 5 million people globally were affected by the 300 major
fire events in the past 30 years, with economic losses of more than US\$ 50 billion (EM-DAT; http://www.emdat.be).
Air quality is regionally affected by the occurrence of fire due to increases in aerosol and ozone that are harmful to
human health. At a regional scale, hospitalisations and human deaths increase in major fire years (Marlier et al.,
2013). The degradation of air quality caused by fire is estimated to result in 260,000 to 600,000 premature deaths
globally each year (Johnston et al., 2012).
Given that fire impacts so many aspects of the earth system, there is considerable concern about what might happen
to fire regimes in response to projected climate changes in the 21$^{st}$ century. However, as the IPCC Fifth Assessment
Report (AR5) made clear, "There is low agreement on whether climate change will cause fires to become more or
less frequent in individual locations" (Settele et al., 2014). This is in large part due to the complexity of the
interactions and feedbacks between vegetation, people, fire and other elements of the earth system (Fig. 1), which is
not well represented in current Earth System Models. Fire, vegetation and climate are intimately linked: changes in
climate drive changes in fire as well as changes in vegetation that provides the fuels for fire, and in return fire alters
vegetation structure and composition, with feedbacks to climate through changing surface albedo, ecosystem
properties, transpiration, and as a source of $CO_2$, other trace gases, and aerosols, altering atmospheric composition
and chemistry (Ward et al., 2012). Human activities strongly affect fire regimes (Archibald et al., 2013) due to the
use of fire for land management, while the use of fire as a tool in the deforestation process is still occurring in the
tropics (e.g. Morton et al., 2008). Humans may also suppress fire directly or indirectly through land-use change
(Bistinas et al., 2014; Knorr et al., 2014; Andela and van der Werf, 2014). Grazing herbivores (the densities of which
are also often controlled by humans) can also decrease fire occurrence by reducing fuel loads (Pachzelt et al. 2015).
Statistical models (e.g. Moritz et al., 2012) have been used to examine the potential trajectory of changes in fire risk,
i.e. the possibility of fire occurring based on climate conditions and fuel availability. Fire risk is not quantitatively
related to area burnt, fuel consumption, or fire emissions. This prevents an assessment of feedbacks to climate
through fire-driven changes of land-surface properties, vegetation structure or atmospheric composition. It is
important to understand such feedbacks quantitatively, as they have the potential to exacerbate or ameliorate the
effects of future climate change on ecosystems, as well as affect the security and well-being of people.
In contrast to statistical models, fire-enabled dynamic global vegetation models (DGVMs) and terrestrial ecosystem
models (TEMs) can address some of the feedbacks between fire and vegetation. Coupling fire-enabled DGVMs with
climate and atmospheric chemistry models in an Earth System Model (ESM) framework allows the feedbacks
between fire and climate to be examined. There has been a rapid development of fire-enabled DGVMs in the past
two decades with many DGVM´s currently including fire as a standard process. Four out of the 15 carbon-cycle
models in the MsTMIP (Multi-scale Synthesis and Terrestrial Model) intercomparison project, 5 out of 10 carbon-



cycle models in TRENDY (Trends in net land-atmosphere carbon exchange over the period 1980-2010), and 9 ESM
in CMIP5 (fifth phase of the Coupled Model Intercomparison Project) provide fire outputs. The complexity of the
fire component of these models varies enormously—from simple empirically-based schemes to predict burnt area,
through models that explicitly simulate the process of ignition and fire spread, to models that incorporate fire
adaptations and their impact on the vegetation response to fire. However, to date there has been no systematic
comparison and evaluation of these models, and thus there is no consensus about the level of complexity required to
model fire and fire-related feedbacks realistically.
The Fire Model Intercomparison Project (FireMIP), initiated in 2014, is a collaboration between fire modelling
groups worldwide to address this issue. Modelling groups participating in FireMIP will run a set of common
experiments to examine fire under present-day and past climate scenarios, and will conduct systematic data-model
comparisons and diagnosis of these simulations with the aim of providing an assessment of the reliability of future
projections of changes in fire occurrence and characteristics. Here, as a background to the FireMIP programme, we
present an overview of the current state of knowledge about the drivers of global fire occurrence. We indicate how
these have been treated over time in different fire models and describe the state-of-the-art fire-enabled DGVMs.
Finally, we outline the FireMIP philosophy and approach to model benchmarking and evaluation.

## 2.      The controls on fire

Fire is driven by complex interactions between climate, vegetation and people (Fig. 1), the importance of which vary
depending on temporal and spatial scales. On meteorological time scales (i.e., minutes to days) and limited spatial
scales (i.e. metres to kilometres), atmospheric circulation patterns and moisture advection determine the location,
incidence and intensity of lightning storms that produce fire ignitions. Weather and vegetation state also determine
surface wind speeds and vapour-pressure gradients, and hence the rates of fuel drying, which in turn affect the
probability of combustion as well as fire spread. However, topography also affects the spread of fire: fire fronts
travel faster uphill because of upward convection of heat while natural barriers such as rivers, lakes, and rocky
outcrops can act as natural barriers to fire fronts.
On longer time scales (i.e., seasons to years) and larger spatial scales (i.e. regional to continental), temperature and
precipitation exert a major effect on fire because these climate variables influence net primary productivity (NPP),
vegetation type and the abundance, composition, moisture content, and structure of fuels. Burnt area tends to be
lowest in very wet or very dry environments, and highest in areas of intermediate water availability. Related to this,
burnt area is greatest at intermediate levels of NPP and decreases with both increases and decreases in productivity.
These unimodal patterns along precipitation or productivity gradients emerge due to the interaction between moisture
availability and productivity: dry areas have low NPP which limits fuel availability and continuity, while NPP and
hence fuel loads are high in wet areas but the available fuel is generally too wet to burn. Temperature exerts an
influence on the rate of fuel drying in addition to its influence on NPP. Seasonality in water availability also plays a
role here: for any given total amount of precipitation, fire is more prevalent in seasonal climates because fuel



accumulates rapidly during the wet season and subsequently dries out. While the vegetation and fuel exert an important control on fire occurrence, fire impacts vegetation distribution and structure, causing important vegetation-fire feedbacks. At a local scale fires create spatial heterogeneity in fuel amount, influencing subsequent fire spread and limiting fire growth.

While natural factors are important drivers of global fire occurrence, human influences are also pervasive. People start fires, either accidentally or with a purpose, for example for forest clearance, agricultural waste burning or fire management. People can also affect fire regimes through land conversion from less flammable (forest) vegetation to more flammable (grassy) vegetation. The introduction of flammable invasive species is another cause of changing fire occurrence. Changes in land use can also reduce fuel loads through crop harvesting, grazing and forestry. Human activities lead to fragmentation of natural vegetation which affects fire spread and fires are also actively suppressed. There is a unimodal statistical relationship between burnt area and population density. At extremely low population densities, increasing population is associated with an increase in fire numbers and burnt area. At high population densities, increasing population is associated with a decrease in burnt area. However, in general when climate and vegetation factors are accounted for, there is a monotonic negative relationship between burnt area and human population, i.e. burned area decreases with increasing human presence (Bistinas et al., 2014; Knorr et al., 2014). The unimodal statistical relationship of burnt area with population density (and other socio-economic variables such as gross domestic product, GDP, that are linked to population density) results from the co-variance of population density with vegetation production and moisture. Low population densities are found in very dry or cold climates where vegetation productivity and fuel loads are also minimal. High population densities are (generally) found in moist environments with high vegetation productivity but where moist conditions limit fire spread.

## 3. History and current status of global fire modelling

While not explicitly representing fire occurrence, early vegetation models often included a generic treatment of disturbance on plant mortality. There are two basic types of fire models that are applied in global vegetation models (Fig. 2): (a) top-down "empirical models" based on statistical relationships between key variables (climate, population density) and some aspect of the fire regime, usually burnt area; and (b) bottom-up "process-based models" which represent small scale fire dynamics (i.e. by simulating individual fires), before scaling up to calculate fire metrics for an entire grid cell. The boundaries between these two types are not rigid, however, and some models combine features of both.

### 3.1 Empirical global fire models

The absence of global-scale fire information before remotely sensed burnt area products became available was a common challenge to the development of fire models and hindered testing and parameterisation of empirical algorithms. The GLOBal FIRe Model (Glob-FIRM) (Thonicke et al., 2001) was the first global fire model, based on the notion that once there is sufficient combustible material burned area depends on the length of the fire season. The



fire season length is calculated as the summed daily "probability of fire" which is a function of the fuel moisture
(approximated by the moisture in the upper soil layer), and the moisture of extinction. The functions relating
moisture content, fire season length, and burnt area were calibrated using site-based observations. In addition, Glob-
FIRM has a threshold value of 200 gC/m$^2$ to represent the point at which fuel becomes discontinuous and the
probability of fire occurring is zero. Glob-FIRM was initially developed for inclusion in the Lund-Potsdam-Jena
(LPJ) DGVM (Sitch et al., 2003), but has since been coupled into several other DGVMs (with some modifications),
including the Common Land Model (Dai et al., 2003), the Community Land Model (CLM) (Levis et al., 2004), the
ORganizing Carbon and Hydrology In Dynamic EcosystEms (ORCHIDEE) (Krinner et al., 2005), the Lund-
Potsdam-Jena General Ecosystem Simulator (LPJ-GUESS) (Smith et al., 2001), and the Biosphere Energy-Transfer
Hydrology model (BETHY) (Kelley, 2008; Kaminski et al., 2013). A simple fire model with a similar structure to
Glob-FIRM, has also been included in the JSBACH global vegetation model (Reick et al., 2013).
Some empirical models include human impacts on fire occurrence. Typically, algorithms are used that link fire
probability/frequency to both an estimate of lightning ignition and to human population density. Pechony and
Shindell (2009) proposed an algorithm whereby the number of fires increases with population, levelling off at
intermediate population densities and then decreasing to mimic fire suppression under high population densities
(Table 1). The simulated number of fire counts are then converted into burnt area using an "expected fire size"
scaling algorithm (Pechony and Shindell, 2009). The human ignition and suppression relationships described by
Pechony and Shindell (2009) have been adopted by several other, both empirical and process based fire-vegetation
models (Table 1). In an alternative approach, Knorr et al. (2014) used a combination of weather information (to
account for fire risk) with remotely-sensed data of vegetation properties that are linked to fire-spread and information
on global population density to derive burned area in a multiple-regression approach. This model has been coupled to
LPJ-GUESS DGVM (Knorr et al., 2016).

### 3.2 Process-based global fire models

MC-FIRE (Lenihan et al., 1998; Lenihan and Bachelet 2015) was the first attempt to simulate fire via an explicit,
process-based, Rate of Spread (RoS) model. MC-FIRE calculates whether a fire occurs in a grid cell on a given day,
based on whether the grid cell is experiencing drought conditions and that the "probability of ignition and spread," as
jointly determined by the moisture of the fine fuel class and the simulated rate of spread, is greater than 50%. The
rate of spread is calculated based on equations by Rothermel (1972), which represent the energy flux from a flaming
front based on fuel size, moisture, and compaction. Canopy fires are initiated using the van Wagner (1993)
equations. All of the grid cell is assumed to burn if a fire occurs, i.e. the original MC-FIRE was designed to simulate
large, intense fires. Later work introduced functions to suppress area burned by low-intensity and/or slow-moving
fires (Rogers et al., 2011). MC-FIRE inspired the development of several process-based RoS based models, and
many fire-enabled DGVMs still use a similar basic framework (Table 1).



The Regional Fire Model (Reg-FIRM: Venevsky et al., 2002) introduced a new approach in fire modelling by
simulating burned area as the product of number of fires and average fire size. Reg-FIRM assumes a constant global
lightning ignition rate, and includes human ignitions depending on population density. It then uses the Nesterov
Index, an empirical relationship between weather and fire, to determine the fraction of ignitions that start fires. Every
fire occurring during a given day in a given grid cell is assumed to have the same properties and thus to be the same
size. Reg-FIRM uses a simplified form of the Rothermel (1972) equations to calculate rate of spread; these
effectively depend only on wind speed, fuel moisture (as approximated by near-surface soil moisture), and PFT-
dependent fuel bulk density. Fire duration is determined stochastically from an exponential distribution with a mean
of 24 hours, to account for the fact that less frequent large fires account for a disproportionate amount of the total
area burned. The RoS equations are used to estimate the burned surface by approximating the shape of the fire as an
ellipse, as suggested by van Wagner (1969).
The fire module in the Canadian Terrestrial Ecosystem Model (CTEM: Arora & Boer, 2005; Melton and Arora,
2015), uses a variant on the Reg-FIRM scheme where the pre-defined FDI approach is replaced by an explicit
calculation of susceptibility, which is the product of the probabilities associated with fuel, moisture, and ignition
constraints on fire (Table 1). Ignitions are either caused by lightning, the incidence of which varies spatially, or
anthropogenic. Anthropogenic ignition is constant in CTEMv1 (Arora & Boer, 2005) but varies with population
density in CTEMv2 (Melton and Arora, 2015). As in Reg-FIRM, fire duration is determined in such a way as to
incorporate the disproportionate area burned by long-lasting fires, but CTEM does this deterministically rather than
stochastically. CTEM includes fire suppression via a "fire extinguishing" probability to account for suppression by
natural and man-made barriers, as well as deliberate human suppression of fires. The fire model development in
CLM (Kloster et al. 2010, and Li et al., 2012; 2013) is based on the CTEM work but introduced anthropogenic
ignitions and suppression on fire occurrence as functions of population density. Li et al. (2013) set anthropogenic
ignitions and suppression also as functions gross domestic production (GDP), and introduced human suppression on
fire spread.
The SPread and InTensity of FIRE (SPITFIRE) model (Table 1) (Thonicke et al., 2010) is a RoS-based fire model
developed within the Lund-Potsdam-Jena (LPJ) DGVM. It is a further development of the Reg-FIRM approach, but
SPITFIRE uses the complete set of physical representations to calculate both rate of spread and fire intensity.
However, maximum fire duration is limited to four hours. Anthropogenic ignitions are a function of population
density as in REGFirm, although the function is regionally tuned in SPITFIRE. Fire is excluded from agricultural
areas but SPITFIRE effectively includes human fire suppression on other lands because human ignitions first
increase and then decrease with increasing population density. The SPITFIRE model has been implemented with
modifications in other DGVM's, including ORCHIDEE (Yue et al., 2014), JSBACH (Lasslop et al., 2014), LPJ-
GUESS (Lehsten et al., 2009), and CLM-ED (Fisher et al., 2014).
Some fire models based on SPITFIRE, such as the Land surface Processes and eXchanges model (LPX) (Prentice et
al., 2011; Kelley et al., 2014) and the Lausanne-Mainz fire model (LMfire) (Pfeiffer et al., 2013), have introduced
further changes to the ignitions scheme. Natural ignition rates in both models are derived from a monthly lightning



climatology, as in SPITFIRE, but LPX preferentially allocates lightning to days with precipitation (which precludes
burning) such that only a realistic number of days have ignition events. Similarly to LPX, LMfire limits lightning
strikes to rain days, and also estimates interannual variability in lightning ignitions by scaling a lightning climatology
using long-term time-series of convective available potential energy (CAPE) produced by atmosphere models.
LMfire further reduces lightning ignitions based on the fraction of land already burnt, since lightning tends to strike
repeatedly in the same parts of the landscape while being rare in others. LPX and LMfire also modified the treatment
of anthropogenic burning relative to the original SPITFIRE. LMfire specified that the number of anthropogenic
ignitions differs amongst livelihoods by distinguishing human populations into three basic categories: hunter-
gatherers, pastoralists, and farmers. Each of these populations has different behaviour with respect to burning based
on assumptions regarding land management goals. LPX, on the other hand, does not include human ignitions on the
grounds that the supposed positive relationship of population density to fire activity is an artefact, as discussed
above. Finally, LMfire accounts for the constraint on fire spread imposed by fragmentation of the burnable landscape
by human land use (as well as topography) while individual fires are allowed to burn across multiple days, and fires
occurring simultaneously within the same grid cell can effectively coalesce as they grow larger.  Like LMfire, the
HESFIRE model (Le Page et al., 2015) also focuses on the constraints on fire spread – using landscape
fragmentation (due to human activities, topography, or past fire events) to determine the probability of extinction of a
fire that is ignited.
Schemes to simulate anthropogenic fire associated explicitly with land-use change have also been developed. Kloster
et al. (2010) include burning associated with land-use change by assuming that some fraction of cleared biomass is
burned. This fraction depends on the probability of fire as mediated by moisture, such that the combusted fraction is
low in wet regions (e.g. northern Europe) and high in dry regions (e.g. central Africa). Li et al. (2013) proposed an
alternative scheme to model fires caused by deforestation in the tropical closed forests, in which fires depended on
deforestation rate and weather/climate conditions, and were allowed to spread beyond land-type conversion regions
when weather/climate conditions are favourable. When the scheme was used in their global fire model, fires due to
human and lightning ignitions described in Li et al. (2012) were not used in the tropical closed forests. Li et al.
(2013) also include cropland management fires, prescribing seasonal timing based on satellite observations but
allowing the amount of burning to depend on the amount of post-harvest waste, population density, and gross
domestic product, and fires in peatlands, depending on a prescribed area fraction of peatland distribution, climate and
area fraction of soil exposed to air.

### 3.3 Modelling the impact of fire on vegetation and emissions

The impact of fire on vegetation operates through combustion of available fuel, plant mortality, and triggering of
post-fire regeneration. There is more similarity in the treatment of fire impacts between models than many other
aspects of fire.



Glob-FIRM assumes that all the aboveground litter/biomass is burnt, while subsequent models assume that only a
fraction of the available fuel is burnt. In CTEM, the completeness of combustion varies by fuel class and PFT (Arora
and Boer, 2005) while models such as MC-FIRE and SPITFIRE include a dynamic scheme for completeness of
combustion which depends on fire characteristics and the moisture content of each fuel class (Thonicke et al., 2010;
Lenihan et al., 1998).
Post-fire vegetation mortality is generally represented in a relatively simple way in fire-enabled DGVMs (Table 2).
Glob-FIRM, CTEM, Reg-FIRM, and the models described by Li et al. (2012) and Kloster et al. (2010) use PFT-
specific parameters for fractional mortality. MC-FIRE has a more explicit treatment of mortality, in which fire
intensity and residence time influence tree mortality from ground fires via crown scorching and cambial damage.
Canopy height relative to flame height (which is a function of fire intensity) determines the extent of crown
scorching. Bark thickness, which scales with tree diameter, protects against damage to the trunk, such that thicker-
barked trees have more chance of surviving a fire of a given residence time. LPJ-SPITFIRE uses a similar approach
except that bark thickness scales with tree diameter, which, together with canopy height depends on woody biomass.
LMfire includes a simple representation of size cohorts within each PFT, with the bark thickness scalar being defined
explicitly for each size cohort. In contrast, gap-based vegetation-fire models such as LPJ-GUESS-
SPITFIRE/SIMFIRE (Lehsten et al. 2009; Knorr et al. 2016) and ED-SPITFIRE (Fisher et al. 2015), explicitly
simulate size cohorts within patches characterised by differential fire-disturbance histories. LPX-Mv1 (Kelley et al.,
2014) incorporates an adaptive bark thickness scheme, in which a range of bark thicknesses is defined for each PFT.
Since thinner-barked trees are more likely to be killed by fire, the distribution of bark thickness within a population
changes in response to fire frequency and intensity.
LPX-Mv1 (Kelley et al., 2014) is the only model to date to incorporate an explicit fire-triggered regeneration
process, through creating resprouting variants of the temperate broad-leaved and tropical broad-leaved tree PFTs.
Resprouting trees are penalised by having low recruitment rates into gaps caused by fire and other disturbances.
However, resprouting is only one part of the syndrome of vegetation responses to fire which include e.g. obligate
seeding, serotiny, and clonal reproduction (e.g. Pausas and Keeley, 2014).

## 4.      Objective and organization of FireMIP

Existing fire models have very different levels of complexity, both with respect to different aspects of the fire regime
within a single model and with respect to different families of models. It is not clear what level of complexity is
appropriate to simulate fire regimes globally. Given the increasing use of fire-enabled DGVMs to project the impacts
of future climate changes on fire regimes and estimate fire-related climate feedbacks (e.g. Knorr et al., 2016; Kelley
and Harrison, 2014; Kloster et al., 2012; Pechony and Shindell, 2010), it is important to address this question.
Coordinated experiments using identical forcings allow comparisons focusing on differences in performance driven
by structural differences between models. The baseline FireMIP simulation will use prescribed climate, $CO_2$,



lightning, population density, and land use forcings from 1700 through 2013. Examination of the simulated
vegetation and fire during the 20$^{th}$ century will allow differences between models to be quantified, and any
systematic differences between types of models or with model complexity to be identified.
However, a single experiment of this type is unlikely to be sufficient to diagnose which processes cause the
differences between models. Various approaches can be used for this purpose, including sensitivity experiments and
parameter-substitution techniques. Similarly, the effect of model complexity can be examined by switching off
specific processes. In FireMIP, experiments will be performed to study the impact of lightning, pre-industrial burned
area, $CO_2$, nitrogen, and fire itself, between different models.
Many model intercomparison projects have shown that model predictions may show reasonably good agreement for
the recent period but then diverge strongly when forced with a projected future climate scenario (e.g. Flato et al.,
2013; Freidlingstein et al., 2014; Harrison et al., 2015). "Out-of-sample" evaluation is one way of identifying
whether good performance under modern conditions is due to the concatenation of process tuning. Within FireMIP,
we will use simulations of fire regimes for different climate conditions in the past (i.e., outside the observational era
used for parameterisation and/or parameter tuning) as a further way of evaluating model performance and the causes
of model-model differences.

**5.       Benchmarking and evaluation in FireMIP**
Evaluation is integral to the development of models. Most studies describing vegetation-model development provide
some assessment of the model's predictive ability by comparison with observations (e.g. Sitch et al., 2003;
Woodward and Lomas, 2004; Prentice et al., 2007). However, these comparisons often focus on the novel aspects of
the model and are largely based on qualitative measures of agreement such as map comparison (e.g. Gerten et al.,
2004; Arora and Boer, 2005; Prentice et al., 2011; Thonicke et al., 2010). However, they often do not track
improvements or degradations in overall model performance caused by these new developments.
The concept of model benchmarking, promoted by the International Land Model Benchmarking Project (ILAMB:
http://www.ilamb. org), is based on the idea of a comprehensive evaluation of multiple aspects of model performance
against a standard set of targets using quantitative metrics. Model benchmarking has multiple functions, including (a)
showing whether processes are represented correctly, (b) discriminating between models and determining which
perform better for specific processes, and (c) making sure that improvements in one part of a model do not
compromise performance in another (Randerson et al., 2009; Luo et al., 2012; Kelley et al., 2013). Since fire affects
many inter-related aspects of ecosystem dynamics and the Earth system, with many interactions being non-linear, the
latter is particularly important for fire modelling.
Kelley et al. (2013) have proposed the most comprehensive vegetation-model benchmarking system to date. This
system provides a quantitative evaluation of multiple simulated vegetation properties, including primary production,





seasonal net ecosystem production, vegetation cover, composition and height, fire regime; and runoff. The
benchmarks are derived from remotely sensed gridded datasets with global coverage, and site-based observations
with sufficient coverage to sample a range of biomes on each continent. Data sets derived using a modelling
approach that involves calculation of vegetation properties from the same driving variables as the models to be
benchmarked are explicitly excluded. The target datasets in the Kelley et al. (2013) scheme allow comparisons of
annual average conditions, seasonal and inter-annual variability. They also allow the impact of spatial and temporal
biases in means and variability to be separately assessed. Specifically designed metrics quantify model performance
for each process, and are compared to scores based on the temporal or spatial mean value of the observations and to
both a "mean" and "random" model produced by bootstrap resampling of the observations.
The Kelley et al. (2013) scheme provides the starting point for model evaluation and benchmarking in FireMIP, but
does not address key aspects of the coupled vegetation-fire system including the amount of above-ground biomass
and/or carbon, fuel load and fuel type, soil and/or fuel moisture, the number of fire starts, fire intensity, the amount
of biomass consumed in individual fires, and fire-related emissions. Global datasets of some of these properties are
now available, including above-ground biomass both derived from vegetation optical depth (Liu et al., 2015) and
ICESAT-GLAS LiDAR data (Saatchi et al., 2011), the European Space Agency Climate Change Initiative Soil
Moisture product (Dorigo et al., 2010), the Global Fire Assimilation System biomass burning fuel consumption
product, fire radiative power, and biomass-burning emissions (Kaiser et al., 2012), and fuel consumption (van
Leeuwen et al., 2014). These will be incorporated into the FireMIP benchmarking scheme. The goal is to provide a
sufficient and robust benchmarking scheme for evaluation of fire while ensuring that other aspects of the vegetation
model can also be evaluated.
The selection of target data sets, in particular how to deal with differences between products and uncertainties, is an
important issue in benchmarking. There are, for example, multiple burnt area products (e.g. GFED4, L3JRC,
MCD45, and ESA MERIS: see Table 3). In addition to the fact that all of these products systematically
underestimate burnt area because of difficulties in detecting small fires (Randerson et al., 2012), they differ from one
another. Although all four products show a similar spatial pattern with more burnt area in the tropical savannas and
less in temperate and boreal regions, L3JRC and MCD45 have a higher total burnt area than MERIS or GFED4
(Table 3). Differences between products are lower (though still substantial) in the tropical savannas than elsewhere;
extra-tropical regions are the major source of uncertainty between products (Fig. 3a). The same is true for interannual
variability (Fig. 3b), where differences between products are higher in regions where total burnt area is low. Most
products show an increase in burnt area between 2001 and 2007 in extra-tropical regions, but there are disagreements
even for the sign of regional changes (Fig. 3c). These types of uncertainties, which are also characteristic of other
data sets, need to be taken into account in model benchmarking—either by focusing on regions or features which are
robust across multiple products or by explicitly incorporating data uncertainties in the benchmark scores (see e.g.
Hargreaves et al., 2013).
Process analyses can provide an alternative approach to model evaluation. The idea here is to identify relationships
between key aspects of a system and potential drivers, based on analysis of observations, and then to determine



whether the model reproduces these relationships (see e.g. Lasslop et al., 2014; Li et al., 2014). It is important to use techniques that isolate the independent role of each potential driving variable because relationships between assumed drivers are not necessarily causally related to the response. Bistinas et al (2014) showed, for example, that burnt area increases as net primary productivity (NPP) increases and decreases as fuel moisture increases. Given that increasing precipitation increases both NPP and fuel moisture this results in a peak in fire at intermediate levels of NPP and precipitation. Population density is also strongly influenced by NPP (i.e. the capacity of the land to provide ecosystem services) and thus the apparent unimodal relationship between burnt area and population density (see e.g. Aldersley et al., 2011) is an artefact of the relationship between population density and NPP. However, when appropriate techniques are used to isolate causal relationships, the ability to reproduce these relationships establishes that the model is simulating the correct response for the right reason. Thus, process-evaluation goes a step beyond benchmarking and assesses the realism of model behaviour rather than simply model response, a very necessary step in establishing confidence in the ability of a model to perform well under substantially different conditions from present.

One goal of FireMIP is to develop modelling capacity to predict the trajectory of fire-regime changes in response to projected future climate and land-use changes. It has been repeatedly shown that vegetation and carbon-cycle models that reproduce modern conditions equally well produce very different responses to future climate change (e.g. Sitch et al., 2008; Friedlingstein et al., 2014). The interval for which we have direct observations is short and does not encompass the range of climate variability expected for the next century. Benchmarking using modern observations does not provide an assessment of whether model performance is likely to be realistic under radically different climate conditions. The climate-modelling community use records of the pre-observational era to assess how well models simulate climates significantly different from the present (Braconnot et al., 2012; Flato et al., 2013; Harrison et al., 2014; Schmidt et al., 2014; Harrison et al., 2015). FireMIP will extend this approach to the evaluation of fire-enabled vegetation models, building on the work of Brücher et al. (2014). Many data sources provide information about past fire regimes. Charcoal records from lake and mire sediments provide information about local changes in fire regimes through time (Power et al., 2010) and have been used to document spatially coherent changes in biomass burnt (Daniau et al., 2012; Marlon et al., 2008; Marlon et al., 2013). Hemispherically-integrated records of vegetation and fire changes can be obtained from records of trace gases (e.g. carbon monoxide), and markers of terrestrial productivity and biomass burning (e.g. carbonyl sulphide, ammonium ion, black carbon, levoglucosan, vanillic acid) in polar ice cores (e.g. Wang et al., 2010; Kawamura et al., 2012; Wang et al., 2012; Asaf et al., 2013; Petrenko et al., 2013; Zennaro et al., 2014). Both hemispherically-integrated and spatially-explicit records of past changes in fire will be used for model evaluation in FireMIP.

## 6.    The next steps

There has been enormous progress in global fire modelling over the past 10–15 years. Knowledge about the drivers of fire has improved, and understanding of fire feedbacks to climate and the response of vegetation is improving.





Global fire models have developed from simulating burnt area only to representing all of the key aspects of the fire
regime. However, there are large and to some extent arbitrary differences in the representation of key processes in
process-based fire models and little is known about the consequences for model performance. While the
development of fire models has been towards increasing complexity, it is still not clear whether a global fire model
*needs* to represent ignition, spread, and extinction explicitly or whether it would be sufficient to just represent the
emergent properties of these processes (burnt area, or fire size, season, intensity, and fire number) in models with
fewer uncertain parameters. The answer to this question may depend on whether the goal is to characterize the role
of fire in the climate system or to understand the interaction between fire and vegetation. Burnt area and biomass are
the key outputs needed to quantify fire frequency and carbon, aerosol and reactive trace gas emissions and changes in
albedo required by climate and/or atmospheric chemistry models. Empirical models may be adequate to estimate
such changes. Other aspects of the fire regime are important factors with respect to the vegetation response to fire
and thus may require a more explicit simulation of e.g. fire intensity and crown fires. By systematically evaluating
models that use different approaches and have different levels of complexity in the treatment of processes in
FireMIP, we hope to acquire new insights to guide future model development.

**7.    Acknowledgments**
S. Hantson and A. Arneth acknowledge support by the EU FP7 projects BACCHUS (grant agreement no. 603445 )
and LUC4C (grant ag. no. 603542).  This work was supported, in part, by the German Federal Ministry of Education
and Research (BMBF), through the Helmholtz Association and its research programme ATMO, and the HGF
Impulse and Networking fund. The MC-FIRE model development was supported by the global change research
programs of the Biological Resources Division of the U.S. Geological Survey (CA 12681901,112- ), the U.S. Dep. of
Energy (LWT-6212306509), the U.S. Forest Service (PNW96--5I0 9 -2-CA), as well as funds from the Joint Fire
Science Program. I. C. Prentice is supported by the AXA Research Fund under the Chair Programe in Biosphere and
Climate Impacts, part of the Imperial College initiative Grand Challenges in Ecosystems and the Environment. F. Li
was funded by the National Natural Science Foundation (grant agreement no. 41475099 and no. 2010CB951801). J.
O. Kaplan was supported by the European Research Council (COEVOLVE 313797).




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



**Tables**

Table 1: Representation of fire processes in fire-enabled DGVM. The intensity of the colour represents the complexity of the description of the process. Shades of grey describe the complexity of the model as a whole: light grey being the simplest; black being the most complex. Blue represents the complexity of description of moisture control on fire susceptibility ranging from: simple statistical relationships/ fire danger indices (FDIs) of fuel as a whole (light blue); description of moisture in multiple fuel size classes; fully modelled or specifically chosen FDIs for specific fuel moisture (dark blue). Green represents the complexity of fuel controlled fire susceptibility: simple masking at a specified fuel threshold (light green); fuel structure effects on ignition probability and rate of spread; and complex modelling of fuel bulk density (dark green). Purple shows complexity of natural ignition schemes: no specified/ assumed ignitions (white); constant ignition source (light purple); simple relationship with fuel moisture; prescribed ignitions - normally through lightning climatology inputs; prescribed lightning with additional scaling for e.g. latitude dependent cloud-ground lightning (CG); daily distributed lightning via a weather generator; and with additional complex ignition simulation (dark purple). Orange represents anthropogenic ignitions: none (white); constant background ignition source (light orange); human population density varying ignitions based on a `human ignition potential' (HIP) and/or gross domestic product (GDP); inclusion of additional, complex human ignition schemes such as pre-historic human behaviour (dark orange). Cyan and lime green represent inclusion of human ignitions suppression and agriculture: none (white); constant suppression (light cyan); increasing suppression with population (medium cyan); simple agricultural masking of fire (light lime green); fuel load manipulation from agriculture (lime green); a mix of agricultural and ignition suppression (dark cyan). Italicize text under `human ignitions' and `human suppression' denote models where the combined influence of human ignitions and suppression result in a unimodal description of fire relative to population density. Brown shows complexity of the calculation of fire sizes, typically through a rate of spread model (RoS): None (white); simplified RoS model to obtain fire properties (light brown); simplified RoS to model individual fires; full Rothermel RoS; multiple RoS models (dark brown). Red show complexity of the calculation of the overall burnt area: the entire cell is affected by fire (light red); constant scaling of the number of fires to burnt area depending on vegetation type; scaling based on moisture and fuel type; entirety of a subcell affected; and scling of number of fires by fire size calculated by RoS model. Arrows demonstrate the exchange of components between models. Arrows start in the model containing the original process description.



| Model | Fuel Moisture | Fuel Load | Fire starts from lightning Ignitions | Anthropogenic Ignitions | Anthropogenic Suppression | Rate of Spread (ROS) | Burnt Area |
|---|---|---|---|---|---|---|---|
| CASA/GFED | None. Fire translated to burnt area from satellite fire counts. | | | | | | Proportional to no. of fires, with more burnt area to fire in sparse vegetation (van der Werf, 2003) |
| GLOBFIRM | Moisture of extinction, above which fire does not occur (Thonicke et al. 2001). Increased fire occurrence with decrease moisture (Thonicke et al. 2001) | Discontinuity fuel load threshold, below which fire does not occur (Thonicke et al. 2001). Reduced fuel from grazing (Krinner et. al. 2005) | | | Suppression from Reduced fuel from grazing (Krinner et al. 2005) | | Increases exponentially with annual (Thonicke et al. 2001) or monthly (Krinner et. al. 2005) summed fire occurrence. |
| SIMFIRE | Maximum possible burnt area a function of FDI (Knorr et al. 2014) | Maximum possible fire as a function of fAPAR as proxie for fuel load (Knorr et al. 2014) | | Increases exponentially with population (Knorr et al. 2014; Knorr et al. 2015) | | | Multiplication of maximum fire functions for fuel, moisture & suppression (Knorr et al. 2014). |
| P&S | Function of VPD (proxy for ambient atmospheric conditions) (Pechony & Shindell, 2009) | Fire scaled by vegetation density based on LAI (Pechony & Shindell, 2009) | Observed lightning flash count, scaled for cloud-to-ground (CG) ratio (Pechony & Shindell, 2009) | | Increases with population (Pechony & Shindell, 2009) | | |
| | | | **Rate of Spread Models** | | | | |
| MC-FIRE | Calculated from fuel size classes and live fuel component (Lenihan et al. 1998). Effects fire start (Lenihan et al. 1998) and RoS (Rothermel 1972) | Size ratios effects RoS (Rothermel 1972) | Fire only occur when 1000hr hour fuel content drops below threshold and rate of spread is above a threshold (Lenihan et al. 1998) | | Capped burnt area for low intensity or slow spread rate fires in populated areas (Rogers et al. 2011) | Fire behaviour scaled by fuel load and moisture based Fire Danger Index (FDI) based rate of spread for ground (Rothermel 1972; Lenihan et al. 1998) and crown (Van Wanger, 1993) fires | Entire grid cell affected by fire during fire occurrence (Lenihan et al. 1998) |
| CTEM | Represented by soil moisture (Arora & Boer 2005; Melton & Arora 2015) | Linear increase fire occurrence between discontinuity and saturated fuel thresholds (Arora & Boar 2005) | Probability of fire occurrence a multiple of probabilities from fuel, moisture & ignitions (Arora & Boar 2005). Latitude dependant CG scaling for lightning (Kloster et al. 2012) | Deforestation fire (Kloster et al. 2012) | No. of days fire burnt suppressed at higher population density (Melton & Arora 2015) | No FDI (Arora & Boar 2005). Affected by differing fuel types (Arora & Boar 2005) | Maximum of 1 fire per sub-grid cell unit. Overall burnt area in grid cell is multiplication of probability of fire by number of units by average fire size per unit (Arora & Boar 2005; Melton & Arora 2015) |
| Li et al. | Represented by soil moisture & relative humidity (Li et al. 2012) | | Ignitions & limitation from fuel and moisture (Li et al., 2012) | Deforestation & degradation fires in tropical closed forests (Li et al. 2013) | Suppression increases with GDP (Li et al. 2013) | | Number of fire multiplied by average area burnt per fire (Venesky et al. 2002) |
| REGFIRM | Fire occurrence from moisture based FDI (Venesky et al. 2002) | | Number of fires instead of probability of fire (Venesky et al. 2002) | 'Human ignition potential'(HPI) (Venesky et al. 2002) | | Variable wind speed affects rate of spread and fire oval shape (Venesky et al. 2002) | |
| SPITFIRE/LPX/Lmfire | | | CG distributed between wet and dry lightning (Prentice et al. 2011). "Storm days" (Kelley et al. 2014). Inter-annual lightning from atmospheric conditions (Pfeiffer et al. 2013) | HIP varying with socieo-economic development (Thonicke et al. 2010). Different human-fire relation for hunter-gatherers, pastoralists and farmers (Pfeiffer et al. 2013) | Cropland fire masking (Thonicke et al. 2010). Additional ignition suppression term (Thonicke et al. 2010). Explicit cropland fragmentation algorithm (Pfeiffer et al. 2013) | Multi-day fires (Pfeiffer et al. 2013). Different RoS for different vegetation type (Pfeiffer et al. 2013). Terrain impediment to spread (Pfeiffer et al. 2013). Reduced rate of spread at high wind speeds (Lasslop et al. 2014) | |

**Legend**

Simple → Complex

Moisture:
- Empirical/FDI base
- Multiple fuel moisture types
- + modeled/ multiple FDI

Fuel:
- Masking threshold
- Fire function of fuel load
- Size classes/ ROS
- Complex

Ignitions:
- Constant/assumed
- Moisture based
- Lightning scaling
- + weather generator
- + complex weather

Anthropogenic:
- Constant
- from pop. density
- deforestation fires
- + additional ignition algorithm

Anthropogenic suppression:
- Agricultural masking
- Fuel manipulation
- Constant suppression
- Varies with pop. density
- + agricultural masking
- + complex masking

Rate of Spread:
- Uses RoS fire properties
- Simplified Rothermel
- Full Rothermel
- Multiple spread types

Burnt Area:
- Entire cell affected
- Simple scaling of no. fires
- Empirically related to fuel and moisture
- Entire sub-cell
- Average burnt area multiplied by no. fires

Relationship



Table 2: Representation of the impacts of fire in fire-enabled DGVMs. Intensity of colour indicates the complexity of
the description of the component. Green indicates complexity of the representation of fire impacts. Red describes the
complexity of the description of atmospheric fluxes from fire: flux is equivalent to all consumed biomass (light red);
consumption based on biomass specific combustion parameters; inclusion of PFT combustion parameters; process
based; biomass/PFT parameterized process-based (dark red). Blue represents the complexity of carbon fluxes to
other carbon pools: no additional fluxes (white); non-combusted dead carbon flux (light blue); carbon fluxes based
on fire spread properties; fire-adapted vegetation carbon retention (dark blue). Orange represents complexity of
simulated mortality processes: parameterized morality (yellow); mortality from crown and cambial damage (light
orange); additional root damage mortality (dark orange). Brown represents complexity of plant adaptation to fire
when mortality processes are included: mortality based on a grid cell's `average plant' properties of fire resistant
traits (light brown); PFT based average traits; inclusion and height cohorts; inclusion of dynamic/complex adaptions
such as resprouting (RS)(dark brown). Arrows demonstrate the exchange of components between models, starting in
the model containing the original description.



| Model (main citation) | Carbon Emission | Other carbon feedbacks | Plant mortality type | Plant resistance |
|---|---|---|---|---|
| CASA/GFED | Combustibility dependent on fuel type (leaf, stem and root, dead) and life-form (wood or grass) *(Potter & Klooster, 1999)* | Killed but not consumed plant material enters litter pool. *(Potter & Klooster, 1999)* | Fraction of woody plants killed dependent on % woody to grass cover. In high wood cover, most trees are killed. Low tree and high grass cover, few trees are killed. *(Potter & Klooster, 1999)* <br> All above-ground grass biomass killed; 90% belowground grass biomass survive *(Potter & Klooster, 1999)* | |
| GLOBFIRM | All aboveground litter & living biomass consumed and released to atmosphere *(Sitch et al. 2003)* | Includes 'Black carbon' (i.e. inert carbon for 1,000s years). *(Krimmer et al. 2005)* | PFT based mortality parameter *(Thonicke et al. 2001)* | |
| **Rate of Spread Models** | | | | |
| MC-FIRE | All canopy carbon is released to atmosphere during crown fires *(Lenihan et al. 1998)* <br> Scorched canopy leafmass from high ground fires released to atmosphere *(Lenihan et al. 1998)* <br> Atmospheric release of consumed dead biomass is calculated from fuel amount and fuel moisture *(Lenihan et al. 1998)* | Scorched woodmass enters litter pool. *(Lenihan et al. 1998)* | Crown scorch mortality based on 'lethal scorch height' of fire and canopy height *(Peterson & Ryan, 2009)* <br> Cambial mortality based on fire residence time and plant bark thickness *(Lenihan et al. 1998)* <br> Root damage *(Lenihan et al. 1998)* | Complete mortality in crown fires *(Lenihan et al. 1998)* <br> Crown/Cambial damage mortality from ground fire follow *Peterson & Ryan (1986)*. All vegetation represented by average crown height and bark thickness, based on simple allometric equations *(Lenihan et al. 1998)* <br> 'Depth of lethal heating' for roots based on *Steward et al. 1990* |
| CTEM | PFT based combustion parameters for different woody components *(Arora & Boar 2005)* | | PFT specific parameters relating carbon consumption to plant mortality *(Arora & Boar 2005)* <br> or PFT-specific mortality factor *(Li et al. 2012)* | |
| REGFIRM | | | | |
| SPITFIRE/ LPX/Lmfire | Fuel load combustion split into PFTs *(Thonicke et al. 2010).* | Carbon retained by surviving resprouting PFTs *(Kelley et al. 2014)* | | Scorch height and bark thickness calculated per PFT, using PFT-specific allometric parameters *(Thonicke et al. 2010).* <br> Within PFT height cohorts affect bark thickness and height-based survival *(Pfeiffer et al. 2013)* <br> Within PFT bark thickness competition *(Kelley et al. 2014)* <br> Resprouting PFTs that resprout from reduced above-ground biomass rather than killed *(Kelley et al. 2014)* |

**Legend:**

Relationship — Simple ↑ Complex ↓

**Carbon combustibility**
- All consumed
- Biomass specific
- + PFT specific
- Process specific
- + PFT/fuel type specific

**Other fluxes**
- Non-combusted carbon -> litter
- Size classes/ROS
- Complex

**Mortality**
- Crown & Cambial
- Crown, Cambial & root kill
- Parameterized mortality

**Survival**
- Based on average plant in grid
- Based on PFT
- + height cohorts
- + Resprouting

Emissions (red arrow) — Carbon pool fluxes (blue arrow) — Mortality process (brown arrow) — Mortality parameters (olive arrow) — Plant resistance (grey arrow)



Table 3: Overview of the burnt area (BA) products used for the intercomparison and their characteristics.

|  | GFED4 | L3JRC | MCD45A1 | ESA MERIS |
|---|---|---|---|---|
| Temporal Resolution | Daily (2001 - present) | Burn date (day) | Burn date (day) | Twice weekly |
| Spatial Resolution | 0.25° | 1km | 500m | 300m |
| Period covered | 1997-present | 2001-2006 | 2001-present | 2006-2008 |
| Mean BA (Mha) | 346.8 | 398.9 | 360.4 | 368.3 |
| Reference | Giglio et al. (2013) | Tansey et al. (2008) | Roy et al. (2008) | Alonso-Canas and Chuvieco (2015) |







**Figures**


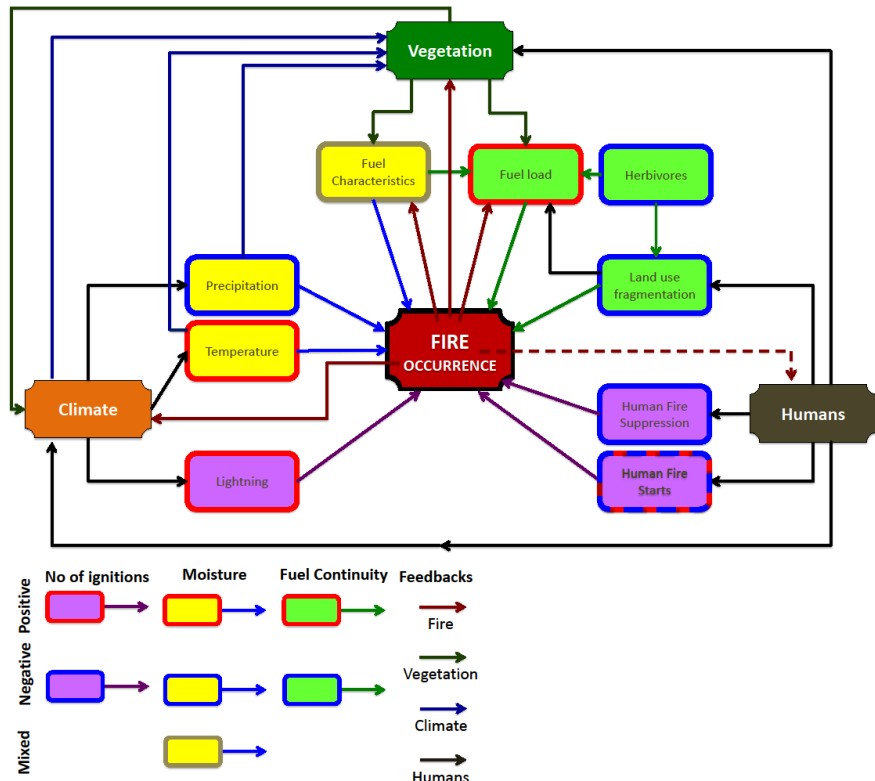


Fig. 1: Summary of the interactions between the controls on fire occurrence on coarse scales. Green boxes show controls influencing fuel; blue influencing moisture; and purple influencing ignitions. Red box indicates positive influence on fire; blue a negative influence, and brown a mixed response. Brown arrows indicate interactions between people and other controls; dark green between vegetation and other controls; and dark blue from climate. Red arrows show feedback from fire.

826

827



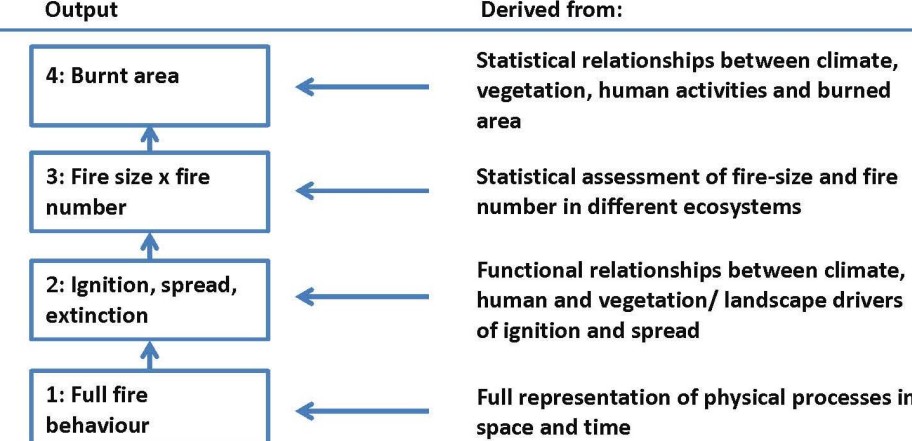

828

Fig. 2: Summarising the levels of model complexity required to derive different aspects of global fire regimes.
Outputs from models functioning at level 1 can be used to derive higher-level outputs, but it is not possible to work
backwards (i.e. empirical relationships between burnt area and environmental drivers will not allow for assessment
of changes in fire number and fire size). Currently there are fire routines in global DGVMs that represent all of these
levels of complexity (see Table 1), and it remains to be decided how much detail is required.











Fig. 3: Coefficient of variation (%) characterizing a) inter-product variability in mean burnt area; b) the inter product
variability of the interannual variability in burned area; and c) the interproduct variability of the slope of temporal
trends (2001-2007). Plots a) and b) are based on all four burnt area products (GFED4, MCD45, L3JRC, ESA
MERIS) whereas plot c) is based on three products and does not include the MERIS data because it is currently only
available for 3 years, see Table 3.