# Peer review of "The status and challenge of global fire modelling"

_Biogeosciences, 2016_

## Referee Comment (RC1) · Anonymous Referee #1 · 2 Feb 2016

General comments:

The paper is suitable for BG, but needs a revision: there are obvious omissions in the paper.

Specific comments:

1.Does the paper address relevant scientific questions within the scope of BG? Yes.

2.Does the paper present novel concepts, ideas, tools, or data? Yes, the paper presents a novel model inter-comparison project.

3.Are substantial conclusions reached? There is no such section as 'Conclusions' in the paper. Therefore, it is not clear whether this paper is aimed to reach some substantial conclusions. It seems that it is focused on "problem statement". "There is a problem", could be interpreted as a substantial conclusion if the problem is formulated clearly and supported by an analysis of the "state-of-the-art". However, authors should

do some effort in this direction. In the present version of the paper, I did not find something that could be considered as a "substantial conclusion", although it is quite obvious for me that the paper may lead to substantial conclusions.

4.Are the scientific methods and assumptions valid and clearly outlined? The scientific methods and assumptions are not clearly outlined, and this make it difficult to judge about their validity. The sections "4. Objective and organization of FireMIP" and "5. Benchmarking and evaluation in FireMIP" are very raw. I would recommend to add a flowchart explaining the conceptual framework of the project objective and organization, and a flowchart explaining the procedure for model benchmarking and evaluation. I also think that authors should address the following questions in the text:

A)Which of the fire modelling groups are eligible to participate in the project? B)Could any group submit its model for benchmarking and evaluation? C)Were all fire modelling groups invited to participate in the project? D)Is the proposed procedure of model benchmarking and evaluation new and original? E)Are the proposed metrics for model benchmarking and evaluation new and original?

5.Are the results sufficient to support the interpretations and conclusions? Yes.

6.Is the description of experiments and calculations sufficiently complete and precise to allow their reproduction by fellow scientists (traceability of results)? Yes.

7.Do the authors give proper credit to related work and clearly indicate their own new/original contribution? I am not sure. It is not clear whether the authors propose new and original procedure and metrics for model benchmarking and evaluation or not.

8.Does the title clearly reflect the contents of the paper? Yes.

9.Does the abstract provide a concise and complete summary? Yes.

10.Is the overall presentation well structured and clear? No. The models are reviewed in somewhat chaotic manner. Some models are not mentioned at all.

11.Is the language fluent and precise? Yes.

12.Are mathematical formulae, symbols, abbreviations, and units correctly defined and used? Yes.

13.Should any parts of the paper (text, formulae, figures, tables) be clarified, reduced, combined, or eliminated? The sections 4 and 5 need major revision.

14.Are the number and quality of references appropriate? No. There are no references to the papers of some active fire modelling groups. For example, I did not find references to the papers recently published by Eliseev:

Eliseev, A.V., I.I. Mokhov, and A.V. Chernokulsky, 2014: An ensemble approach to simulate $CO_2$ emissions from natural fires. Biogeosciences, 11 (12), 3205-3223, doi 10.5194/bg-11-3205-2014

Eliseev, A.V., I.I. Mokhov, and A.V. Chernokulsky, 2014: Influence of ground and peat fires on $CO_2$ emissions into the atmosphere. Doklady Earth Sci., v. 459, no. 2, p. 1565-1569, doi 10.1134/S1028334X14120034

Moreover, there is no one reference on the lines 136-157 where authors review physical controls of fires. This looks strange.

---

## Referee Comment (RC2) · E. Chuvieco (Referee) · 21 Mar 2016

General evaluation The paper presents a framework for evaluation of fire models, particularly in the context of analyzing the potential impacts of climate changes in altering fire regimes. The paper is well written and generally well documented (few minor issues are commented later). It does provide a good overview of existing global DGVM including a fire component, with tables summarizing the main assumptions and drivers. I think the paper would benefit from extending the specific benchmark test that will be used to compare the model performance, as the current version only gives insights on potential approaches. I believe the authors should be more specific on what are the planning to do to actually compare model performance, which indicators will they use, on which period and area (including target resolution).

Minor issues: Line 65: a comma is missed after seasonality: "frequency, intensity, seasonality etc". Line 71: What a significant fraction means? Please quantify Lines 70-

86: No reference to N2O emissions from fires is made. Why? Line 92: Johnston et al., 2012 is missed in the references section. Lines 108-109: "Fire risk is not quantitatively related to area burnt, fuel consumption, or fire emissions". It is not clear what you mean here. Most fire risk systems are assessed with fire statistics (Chuvieco et al. 2014; Chuvieco et al. 2010; Padilla and Vega-Garcia 2011; Paz et al. 2011), and some are associated to burned area and fuel consumption (Consume, for instance: see Pettinari and Chuvieco 2015). Line 143: "outcrops can act as natural barriers to fire fronts". Natural barriers is duplicated from previous line. Line 147: "and highest in areas of intermediate water availability", assuming a dry period exists. Line 159: "purpose, for example for forest clearance, agricultural waste burning or fire", please add pasture management, which is the most common factor in many areas of the world. Line 170: "gross domestic product, GDP, that are linked to population density) results from the co-variance of population density with vegetation production and moisture". This sentence may be tinged, as those relations depend on other factors, such as the importance of agricultural sector in regional economy. For a global analysis, you may be interested to read Chuvieco and Justice 2010. Bowman et al. 2011 has also an interesting analysis of human-fire relations. Line 198: the JSBACH acronym is not defined. Lines 368: When citing alternative sources of model assessment, you do not include reference to the GFED dataset (Giglio et al. 2013), which is widely used for fire –emissions analysis. A reference to the synthesis analysis of Mouillot et al (2014) may be relevant in this point. Please, also note that soil moisture is not equivalent to fuel moisture. The CCI Soil moisture product does not really estimate vegetation wetness. Lines 383-388: When analyzing different global burned area products, you may refer to the intercomparison analysis published by Chang and Song 2009 or the most recent validation effort by Padilla et al. 2015. Line 381 and 814: Please not that ESA MERIS burned area product is officially named Fire_cci (see Chuvieco et al. 2016, which also includes an assessment of fire emissions derived from this product). The temporal resolution of the Fire_cci product is Burn Date for the pixel product at aprox. 300 m resolution. However, the burned area is accumulated in 15 day periods for a gridded

version of product, which has 0.5 d resolution. Line 459: please, include the updated reference to Alonso-Canas and Chuvieco 2015. Line 819. In fig. 1 you may add to Fuel load, Fuel continuity, which is related to fragmentation.

References Alonso-Canas, I., & Chuvieco, E. (2015). Global Burned Area Mapping from ENVISAT-MERIS data Remote Sensing of Environment, 163, 140-152. Bowman, D.M.J.S., Balch, J., Artaxo, P., Bond, W.J., Cochrane, M.A., D'Antonio, C.M., DeFries, R., Johnston, F.H., Keeley, J.E., Krawchuk, M.A., Kull, C.A., Mack, M., Moritz, M.A., Pyne, S., Roos, C.I., Scott, A.C., Sodhi, N.S., & Swetnam, T.W. (2011). The human dimension of fire regimes on Earth. Journal of Biogeography, 38, 2223-2236. Chang, D., & Song, Y. (2009). Comparison of L3JRC and MODIS global burned area products from 2000 to 2007. Journal of Geophysical Research, 114, 10.1029/2008JD11361. Chuvieco, E., Aguado, I., Jurdao, S., Pettinari, M.L., Yebra, M., Salas, J., Hantson, S., de la Riva, J., Ibarra, P., Rodrigues, M., Echeverría, M., Azqueta, D., Román, M.V., Bastarrika, A., Martínez, S., Recondo, C., Zapico, E., & Martínez-Vega, F.J. (2014). Integrating geospatial information into fire risk assessment. International Journal of Wildland Fire, 23, 606–619. Chuvieco, E., Aguado, I., Yebra, M., Nieto, H., Salas, J., Martín, P., Vilar, L., Martínez, J., Martín, S., Ibarra, P., de la Riva, J., Baeza, J., Rodríguez, F., Molina, J.R., Herrera, M.A., & Zamora, R. (2010). Development of a framework for fire risk assessment using remote sensing and geographic information system technologies. Ecological Modelling, 221, 46-58. Chuvieco, E., & Justice, C. (2010). Relations between Human Factors and Global Fire Activity. In E. Chuvieco, J. Li & X. Yang (Eds.), Advances in earth observation of global change (pp. 187-200). Dordrecht ; London: Springer. Chuvieco, E., Yue, C., Heil, A., Mouillot, F., Alonso-Canas, I., Padilla, M., Pereira, J.M., Oom, D., & Tansey, K. (2016). A new global burned area product for climate assessment of fire impacts. Global Ecology and Biogeography, DOI 10.1111/geb.12440. Giglio, L., Randerson, J.T., & Werf, G.R. (2013). Analysis of daily, monthly, and annual burned area using the fourth generation global fire emissions database (GFED4). Journal of Geophysical Research: Biogeosciences, 118, 317-328. Mouillot, F., Schultz, M.G., Yue, C., Cadule, P., Tansey, K., Ciais, P.,

& Chuvieco, E. (2014). Ten years of global burned area products from spaceborne remote sensing—A review: Analysis of user needs and recommendations for future developments. International Journal of Applied Earth Observation and Geoinformation, 26, 64-79. Padilla, M., Stehman, S.V., Hantson, S., Oliva, P., Alonso-Canas, I., Bradley, A., Tansey, K., Mota, B., Pereira, J.M., & Chuvieco, E. (2015). Comparing the Accuracies of Remote Sensing Global Burned Area Products using Stratified Random Sampling and Estimation. Remote Sensing of Environment, 160, 114-121. Padilla, M., & Vega-Garcia, C. (2011). On the comparative importance of fire danger rating indices and their integration with spatial and temporal variables for predicting daily human-caused fire occurrences in Spain. International Journal of Wildland Fire, 20, 1-13. Paz, S., Carmel, Y., Jahshan, F., & Shoshany, M. (2011). Post-fire analysis of pre-fire mapping of fire-risk: A recent case study from Mt. Carmel (Israel). Forest Ecology and Management, 262, 1184-1188. Pettinari, M.L., & Chuvieco, E. (2015). Generation of a global fuel dataset using the Fuel Characteristic Classification System. Biogeosciences Discuss., 12, 17245-17284.
* * *

---

## Author Response (AR1)

**Response to reviewer 1**

*Response: We thank the reviewer for his/her constructive comments. Here we only list review comments that need to be addressed in the revised version (i.e. not comments that were answered positively). The line numbers indicated refer to the track-change version of the manuscript given below.*

Specific comments:

3. There is no such section as 'Conclusions' in the paper. Therefore, it is not clear whether this paper is aimed to reach some substantial conclusions. It seems that it is focused on "problem statement". "There is a problem", could be interpreted as a substantial conclusion if the problem is formulated clearly and supported by an analysis of the "state-of-the-art". However, authors should do some effort in this direction. In the present version of the paper, I did not find something that could be considered as a "substantial conclusion", although it is quite obvious for me that the paper may lead to substantial conclusions.

*Response: The reviewer is correct that this paper is intended to be a statement of a problem, supported by an analysis of the state-of-the-art. Succinctly, we need to be able to predict how fire regimes will change in the future and in order to have confidence in these projections we need to evaluate how well different models perform, and to devise ways of improving model performance if this is necessary, and we have set up the FireMIP in order to make some progress with these goals. We are sorry that this message (or conclusion) did not come through clearly enough. We have changed the title of the final section from "the next steps" to "Conclusions and Next Steps" and expanded the text to strengthen this important point. Specifically, we have added a first paragraph to this section to emphasize that the goal of FireMIP is to demonstrate whether existing fire models are sufficiently mature to be used for projections as follows (Lines 468-473):*

*"Fire has profound impacts on many aspects of the Earth system. We therefore need to be able to predict how fire regimes will change in the future. Projections based on statistical relationships are not adequate for projections of longer-term changes in fire regimes because they neglect potential changes in the interactions between climate, vegetation and fire. While mechanistic modelling of the coupled vegetation-fire system should provide a way forward, it is still necessary to demonstrate that they are sufficiently mature to provide reliable projections. This is a major goal of the FireMIP project."*

*The existing text for this section emphasized the different levels of complexity of existing fire models and the fact that we do not yet know what level of complexity is required to achieve robust results. We have preserved this paragraph, but have revised the final part of this paragraph to point out that another major goal of FireMIP is to establish the level of complexity required as follows (lines 488-492):*

*"FireMIP will address these issues by systematically evaluating the performance of models that use different approaches and have different levels of complexity in the treatment of processes, in order to establish whether there are aspects of simulating modern and/or future fire regimes that require complex models. Systematic evaluation will also help guide future development of individual models and potentially the further development of vegetation-fire models in general."*

*We have used the opportunity of expanding this conclusion section to add text to address the issues about the nature of the FireMIP project raised below (the additional text is given in our response to these questions).*

4. The scientific methods and assumptions are not clearly outlined, and this make it difficult to judge about their validity. The sections "4. Objective and organization of FireMIP" and "5. Benchmarking and evaluation in FireMIP" are very raw. I would recommend to add a flowchart explaining the conceptual framework of the project objective and organization, and a flowchart explaining the procedure for model benchmarking and evaluation.

*Response: Our goal in these sections was to describe the conceptual framework of FireMIP and of the benchmarking that will be performed; we did not want to provide a detailed protocol for the experiments or the benchmarking because these will both be developed during the project itself. We agree that it will be helpful to provide flowcharts for the experiments and for the benchmarking when we document these protocols. However, we think these are not appropriate in a paper that focuses on giving an overview of fire model development and presenting the state-of-the-art in global fire modelling, as an introduction to the need for a model intercomparison project and for benchmarking current models. We have now tried to make the aim and objective of the current manuscript clearer in the abstract and introduction.*

I also think that authors should address the following questions in the text: A) Which of the fire modelling groups are eligible to participate in the project? B) Could any group submit its model for benchmarking and evaluation? C) Were all fire modelling groups invited to participate in the project?

*Response: FireMIP is a community initiative rather than a funded project, and has come about through interactions between a large group of fire-modelling groups worldwide. However, participation in this initiative is open to all fire modelling groups, and also to fire scientists who wish to participate in model analysis. One of the purposes of this manuscript is to advertise the FireMIP project to the wider community, to encourage participation. We welcome the chance to make this clear and have added some text in the final section as follows (Lines 494-497):*

*"FireMIP is a non-funded initiative of the fire-modelling community. Participation in the development of benchmarking data sets and analytical tools, as well as in the running and analysis of the model experiments, is open to all fire scientists. We hope that will maximise exchange of information between modelling groups and facilitate rapid progress in this area of science."*

*We have also taken the opportunity to add an invitation to participate in FIREMIP in the acknowledgement section.*

D) Is the proposed procedure of model benchmarking and evaluation new and original? E) Are the proposed metrics for model benchmarking and evaluation new and original?

*We are sorry that it was not clear that FireMIP will be the first time that benchmarking and evaluation of fire models across standard experiments is carried out. Our benchmarking data is still under development, but as we point out in the manuscript (see also response to reviewer no. 2), the intention is to go well beyond the data sets and metrics that were proposed in Kelley et al. (2013) because we need to have information about other aspects of the fire regime. We have taken the opportunity to make the novelty of the FireMIP initiative clearer by adding a sentence in the last paragraph of the introduction as follows (Lines 139-141):*

*"There has been no previous attempt to compare fire models across a suite of standardised experiments (model-model comparison) or to systematically evaluate model performance using a wide range of different benchmarks (data-model comparison)."*

*In addition, we also clarified some of these aspects in revisions to the benchmarking section (e.g., lines 393-402, and 405-412), including a new paragraph (lines 413-418):*

*"The FireMIP benchmarking system will represent a substantial step forward in model evaluation. Nevertheless there are a number of issues that will need to be addressed as the project develops, specifically how to deal with the existence of multiple data sets for the same variable, how to exploit process understanding in model evaluation, and how to ensure that models which are tuned for modern conditions can respond to large changes in forcing. The answers to these questions remain unclear, but here we provide insights into the nature of the problem and suggest some potential ways forward."*

7.Do the authors give proper credit to related work and clearly indicate their own new/original contribution? I am not sure. It is not clear whether the authors propose new and original procedure and metrics for model benchmarking and evaluation or not.

> *Response: Please see response to point above.*

10.Is the overall presentation well structured and clear? No. The models are reviewed in somewhat chaotic manner. Some models are not mentioned at all.

> *Response: Fire models have developed in parallel to one another, but there has been some overlap between the approaches taken by different models to representing key processes, which has been the logic behind the current structure in presenting the different models. Indeed (as we show in Tables 1 and 2), some process-descriptions have been adopted by several models –either with minor modification or with tuning because of being coupled to different representations of other aspects of the fire regime. For example, many modelling groups have adopted the human ignition and suppression algorithm, and although the population density thresholds used differ, there is nothing fundamental that distinguishes these treatments. Our goal here was not to describe every single fire model in detail, but rather to outline the major approaches to key processes and in particular to mention models when they introduce fundamentally new approaches (which we now have clarified in lines 197-200). We now have included a mention of the fire module in the IAP RAS CM and citing the Eliseev paper as suggested and have also include reference to two more recent fire model developments.*

13.Should any parts of the paper (text, formulae, figures, tables) be clarified, reduced, combined, or eliminated? The sections 4 and 5 need major revision.

> *We have now tried to explain better the objective of the manuscript in the abstract and introduction. We have also adapted the section 5 and the section conclusions and next steps. See also response to comment about sections 4 and 5 above.*

14.Are the number and quality of references appropriate? No. There are no references to the papers of some active fire modelling groups. For example, I did not find references to the papers recently published by Eliseev:
Eliseev, A.V., I.I. Mokhov, and A.V. Chernokulsky, 2014: An ensemble approach to simulate CO2 emissions from natural fires. Biogeosciences, 11 (12), 3205-3223, doi 10.5194/bg-11-3205-2014
Eliseev, A.V., I.I. Mokhov, and A.V. Chernokulsky, 2014: Influence of ground and peat fires on CO2 emissions into the atmosphere. Doklady Earth Sci., v. 459, no. 2, p. 1565-1569, doi 10.1134/S1028334X14120034.

> *Response: We have now included reference to the fire development in IAP RAS CM as well as two other recent fire developments.*

Moreover, there is no one reference on the lines 136-157 where authors review physical controls of fires. This looks strange.

> *Response: The text describing the physical controls on fire basically summarises what is now "common knowledge" in fire science, and various versions of this description appear in all the major reviews published in the last few years. While we feel it is important to provide this context, there is nothing surprising about this section of text. We could provide multiple references for each statement, but there would be little justification for citing one set of papers over others and any choice (to keep the reference list within reasonable limits) would be arbitrary. For this reason, we prefer not to include references to this section of the text.*

**Response to reviewer 2**

*Response: We thank the reviewer for these positive and constructive comments. Below we give a point by point response. The line numbers indicated refer to the track-change version of the manuscript given below.*

General evaluation: The paper presents a framework for evaluation of fire models, particularly in the context of analyzing the potential impacts of climate changes in altering fire regimes. The paper is well written and generally well documented (few minor issues are commented later). It does provide a good overview of existing global DGVM including a fire component, with tables summarizing the main assumptions and drivers. I think the paper would benefit from extending the specific benchmark test that will be used to compare the model performance, as the current version only gives insights on potential approaches. I believe the authors should be more specific on what are the planning to do to actually compare model performance, which indicators will they use, on which period and area (including target resolution).

*Response: Our intention in this paper was to describe the current status of global fire modelling, including the current status of model evaluation, and indicate the challenges which need to be addressed in a modelling intercomparison, rather than providing a detailed protocol for FIREMIP itself and we attempt to make our aims clearer in the revised manuscript. We agree that our description of the different steps involved in developing the FIREMIP evaluation scheme was perhaps not spelled out clearly enough, and that there are details the could have been added. As similar types of requests were also asked by reviewer #1 we have therefore:*

*modified the paragraph describing the Kelley et al. benchmarking system to indicate that this is what we will use initially in FIREMIP. The data sets involved, which provide benchmarks for multiple aspects of both vegetation and fire, were already listed in the paragraph. However we have now also indicated that the comparisons will be made at* a 0.5° *resolution – which is the common grid of the data sets in the Kelley et al system – but that spatial resolution does not have any impact on the metrics. If the comparisons were made at the model resolution, the metric score would be identical with two significant figures. However, given that the models all have different resolutions, and that the benchmark data sets are already at 0.5 ° resolution, the most convenient approach is to refer everything to a common framework. The end of this paragraph now reads (lines 393-396):*

*"The Kelley et al. (2013) scheme will be used for model evaluation and benchmarking in FireMIP. It has been shown that spatial resolution has no significant impact on the metric scores for any of the targets (Harrison and Kelley, unpublished data); nevertheless, model outputs will be interpolated to the 0.5° common grid of the data sets for convenience."*

*We have modified the second paragraph to make it clearer that our intention in FireMIP is to expand the Kelley et al benchmarking system, and in particular to take the opportunity to include new data sets as they appear. This paragraph now reads (lines 397-412):*

*"The Kelley et al. (2013) scheme does not address key aspects of the coupled vegetation-fire system including the amount of above-ground biomass and/or carbon, fuel load, soil moisture, fuel moisture, the number of fire starts, fire intensity, the amount of biomass consumed in individual fires, and fire-related emissions. Global datasets describing some of these properties are now available, and will be included in the FireMIP benchmarking scheme. These data sets include above-ground biomass both derived from vegetation optical depth (Liu et al., 2015) and ICESAT-GLAS LiDAR data (Saatchi et al., 2011), the European Space Agency Climate Change Initiative Soil Moisture product (Dorigo et al., 2010), the Global Fire Assimilation System biomass burning fuel consumption product, fire radiative power, and biomass-burning emissions (Kaiser et al., 2012), and fuel consumption (van Leeuwen et al., 2014). The goal is to provide a sufficient and robust benchmarking scheme for evaluation of fire while ensuring that other aspects of the vegetation model can also be evaluated, and to this end new data sets will be incorporated into the FireMIP benchmarking scheme as they become available during the project."*

*Finally, we have tried to make it clearer that the development of other aspects of the FireMIP evaluation and benchmarking exercise are research questions that will need to be addressed during the project. We are convinced that new approaches are required to deal with uncertainties caused by the fact that different, and apparently equally robust, data sets show substantially different patterns. But the techniques for propagating these uncertainties into metrics are in their infancy. We also believe that process evaluation and palaeo evaluation are necessary steps in model evaluation, but have not been used in any systematic way for fire modelling. Therefore, we have also added the following text before the final three paragraphs in this section (lines 413-418 (see response to reviewer no. 1)) as follows:*

*"The FireMIP benchmarking system will represent a substantial step forward in model evaluation. Nevertheless there are a number of issues that will need to be addressed as the project develops, specifically how to deal with the existence of multiple data sets for the same variable, how to exploit process understanding in model evaluation, and how to ensure that models which are tuned for modern conditions can respond to large changes in forcing. The answers to these questions remain unclear, but here we provide insights into the nature of the problem and suggest some potential ways forward."*

Minor comments:

*Response: All suggestions have been taken into account in the revised version.*

Line 65: a comma is missed after seasonality: "frequency, intensity, seasonality etc".

*Response: corrected*

Line 71: What a significant fraction means? Please quantify

*Response: The current assumption in carbon budgeting is that all of the carbon lost in fires will be taken up as vegetation regrowth within a decade. This does not hold under a changing climate or if people use the post-fire opportunity to convert the area to e.g. crops. Thus, it is difficult to know how to quantify this accurately and indeed no one has done this. We agree that this perhaps deserves fuller treatment as so we have modified the sentence to read (lines 71-74):*

*"This is equivalent to ca 25% of those from fossil fuel combustion (Ciais et al., 2013; Boden et al., 2013), although in the absence of climate and/or land use change, nearly all of these emissions are taken up during vegetation regrowth after fire."*

Lines 70- 86: No reference to N2O emissions from fires is made. Why?

*Response: We had lumped N2O in "many other atmospheric constituents" (Line 75), but agree that it is sufficiently important to mention explicitly and we have now done so.*

Line 92: Johnston et al., 2012 is missed in the references section.

*Response: Added*

Lines 108-109: "Fire risk is not quantitatively related to area burnt, fuel consumption, or fire emissions". It is not clear what you mean here. Most fire risk systems are assessed with fire statistics (Chuvieco et al. 2014; Chuvieco et al. 2010; Padilla and Vega-Garcia 2011; Paz et al. 2011), and some are associated to burned area and fuel consumption (Consume, for instance: see Pettinari and Chuvieco 2015).

*Response: We agree that this text is not clear. It is true that fire risk (or so-called fire danger) systems are developed and assessed based on fire statistics, and often using burned area or fuel consumption as a target. Our point here is that these systems are calibrated under current conditions, but have been used to assess what might happen in the future. There are many papers that do this, and we have focused on the Moritz et al. (2012) paper as an example because this formed the basis of the statements about future fire in the last IPCC assessment. But statistical fire risk/danger models cannot account for a number of factors that could influence future fire regimes, such as the impact of CO2 fertilization on in situ productivity or changes in vegetation type. They also cannot account for the possibility that future climates may not have analogues in the modern day, e.g. because of changes in temperature seasonality. Our point here was to make the case for process-based fire modelling if the goal is to project potential changes in fire regimes in the future. We have rewritten this paragraph to make the argument more explicit as follows (lines 110-120) :*

*"Statistical models have been used to examine the potential trajectory of changes in fire during the 21st century (e.g. Moritz et al., 2012; Settele et al., 2014). Such models essentially assess the possibility of fire occurring given climate conditions and fuel availability (fire risk or fire danger) based on modern day relationships between climate, fuel and some aspect of the fire regime such as burnt area. However, changes in fire risk/danger will not necessarily be closely coupled to changes in fire regime in the future given the direct impacts of CO2 on water-use efficiency, productivity, vegetation density and ultimately vegetation distribution. This limits the utility of statistically-based models for the investigation of feedbacks to climate through fire-driven changes of land-surface properties, vegetation structure or atmospheric composition – feedbacks which have the potential to exacerbate or ameliorate the effects of future climate change on ecosystems, as well as influence the security and well-being of people."*

Line 143: "outcrops can act as natural barriers to fire fronts". Natural barriers is duplicated from previous line.

*Response: We have deleted the duplicated words.*

Line 147: "and highest in areas of intermediate water availability", assuming a dry period exists.

*Response: This statement is globally true regardless of seasonality of precipitation. The highest burnt areas are found in areas with sufficient rainfall to produce good vegetation cover and hence fuel to burn but where rainfall is not so high as to ensure that the fuel is permanently wet. Most such areas do have a marked seasonal cycle of precipitation but this is not necessary to the argument although the timing and length of the dry season affects the quantitative level of what is meant by "intermediate" water availability. We feel that the term "water availability" is somewhat confusing here, and so we have modified the text to make it clearer what we mean as follows:*

*"Burnt area tends to be lowest in very wet or very dry environments, and highest where the water balance is intermediate between these two states."*

Line 159: "purpose, for example for forest clearance, agricultural waste burning or fire", please add pasture management, which is the most common factor in many areas of the world.

*Response: We have added this.*

Line 170: "gross domestic product, GDP, that are linked to population density) results from the co-variance of population density with vegetation production and moisture". This sentence may be tinged, as those relations depend on other factors, such as the importance of agricultural sector in regional economy. For a global analysis, you may be interested to read Chuvieco and Justice 2010. Bowman et al. 2011 has also an interesting analysis of human-fire relations.

*Response: There have indeed been many papers analysing the relationships between postulated drivers of fire and burnt area, both at a regional scale and at a global scale, but many have not taken into account the co-variance between different explanatory variables. Here, our statement is based on the comprehensive global analysis by Bistinas et al. which used GLM to identify the independent relationships between burnt area and specific driver, and showed that spatial and temporal trends in burnt area could be predicted with a simple model based primarily on vegetation productivity and moisture. Bistinas et al. also showed that the relationship between burnt area and population density was significant but negative, and that the unimodal relationships with population and GDP can be reproduced by this simple model. We have added a further reference to the Bistinas et al. (2014) paper to clarify that this is the source for our assertion. We have now included a reference to Bowman et al., 2011 in the introduction.*

Line 198: the JSBACH acronym is not defined.

*We have added the full name of JSBACH, which is the "Jena Scheme for Biosphere-Atmosphere Coupling in Hamburg" at line 198. Thank you for pointing out that this was not defined; the full name is so rarely used that we suspect only the originator of the acronym remembers what it stands for.*

Lines 368: When citing alternative sources of model assessment, you do not include reference to the GFED dataset (Giglio et al. 2013), which is widely used for fire –emissions analysis. A reference to the synthesis analysis of Mouillot et al (2014) may be relevant in this point. Please, also note that soil moisture is not equivalent to fuel moisture. The CCI Soil moisture product does not really estimate vegetation wetness.

*Response: We agree that GFED is by far the most widely used "reference" data set when analysing fire emissions under present day conditions. However, the emissions are calculated using the CASA vegetation model and are therefore not an independent reference data set. As we have stressed in describing the benchmarking system, we have chosen data sets that are not dependent on a model driven by the same drivers as the models we are seeking to test (line 387-389). Thus while we will use GFED burnt area, we will not use GFED emissions for model evaluation. We do not claim that soil moisture is equivalent to fuel moisture (we now separate them in the text to make this clearer). In describing the benchmarking system we have made the point that it is important to evaluate the simulation of both vegetation properties and fire regimes – it may well be that the failure to capture fire regimes is related to under or over production of woody vegetation, for e.g., which is directly related to the simulated soil moisture. However, clearly these points are worth stressing and we have modified the text describing the alternative data sets to emphasise these two points, as follows (lines 406-412):*

*"The selection of new data sets is partly opportunistic, but reflects the need both to evaluate all aspects of the coupled vegetation-fire system and the importance of using data sets that are derived independently of any vegetation model that uses the same driving variables as the coupled vegetation-fire models being benchmarked. The goal is to provide a sufficient and robust benchmarking scheme for evaluation of fire while ensuring that other aspects of the vegetation model can also be evaluated, and to this end new data sets will be incorporated into the FireMIP benchmarking scheme as they become available during the project."*

Lines 383-388: When analyzing different global burned area products, you may refer to the intercomparison analysis published by Chang and Song 2009 or the most recent validation effort by Padilla et al. 2015. Line 381 and 814: Please not that ESA MERIS burned area product is officially named Fire_cci (see Chuvieco et al. 2016, which also includes an assessment of fire emissions derived from this product). The temporal resolution of the Fire_cci product is Burn Date for the pixel product at aprox. 300 m resolution. However, the burned area is accumulated in 15 day periods for a gridded version of product, which has 0.5 d resolution.

*Response: We included the Padilla reference and changed the name of the ESA MERIS product to Fire_cci as indicated. We now indicate that the spatial resolution of MERIS is ± 300m in table 3 and changed the name of the product there, as well as the temporal resolution.*

Line 459: please, include the updated reference to Alonso-Canas and Chuvieco 2015.

*Response: Changed*

Line 819. In fig. 1 you may add to Fuel load, Fuel continuity, which is related to fragmentation.

*Response: We have tried to indicate the role of fuel continuity with the arrow going from fragmentation to fuel load. We now mention this in the header of figure 1.*

**List of the main changes made**

1.  We have adapted the abstract to indicate more clearly what will be presented in the manuscript
2.  We have clarified some points indicated by the reviewers in the introduction.
3.  We have rewritten the paragraph in the introduction on the use of statistical models for future fire projections.
4.  We indicate now in the introduction that there have been no previous attempts to compare and benchmark fire models.
5.  The final paragraph of the introduction is adapted to indicate better the content of the manuscript.
6.  Some minor changes have been performed in section 2, based on the reviewers comments.
7.  In section 3 we now include some lines indicating the objective of the review. We also include now reference to three more fire models.
8.  Various paragraphs of the benchmarking section (5) have been added or rewritten, mainly covering the spatial resolution, the selection of benchmarking datasets and the novelty of the benchmarking approach.
9.  Section 6 (conclusions and next steps) has been strongly adapted, now indicating and invitation to all fire modelling groups, expressing better the reason why we started FireMIP and what we might learn from FireMIP.
10. The bibliography has been updated and the format homogenised.
11. We have included a formal invitation in the acknowledgements for fire modelling groups to participate and contact the first author.

[revised manuscript text omitted]
 coupled land surface vegetation fire model without climate envelopes, CLM4.5(, clm edED), Geosci. Model Dev., 8, 3593-3619, 10.5194/gmd-8-3593-2015, 2015 spitfire., 2014.

Flato, G., Marotzke, J., Abiodun, B., Braconnot, P., Chou, S., Collins, W., Cox, P., Driouech, F., Emori, S., and Eyring, Forest, V.C., Gleckler, P., Guilyardi, E., Jakob, C., Kattsov, V., Reason, C. and Rummukainen, M.: Evaluation of climate models, Inin: Climate Changechange 2013: The Physical Science Basis.physical science basis. Contribution of Working Group Iworking group i to the Fifth Assessment Reportfifth assessment report of the Intergovernmental Panelintergovernmental 
[revised manuscript text omitted]

Carbon pool fluxes

**Mortality**
- Crown & Cambial
- Crown, Cambial & root kill

Mortality process

Parameterized mortality

Mortality parameters

**Survival**
- Based on average plant in grid
- Based on PFT
  - + height cohorts
  - + Resprouting

Plant resistance

Emissions

Table 3: Overview of the burnt area (BA) products used for the intercomparison and their characteristics.

| | GFED4 | L3JRC | MCD45A1 | Fire_cciESA MERIS |
|---|---|---|---|---|
| Temporal Resolution | Daily (2001 - present) | Burn date (day) | Burn date (day) | Burn date (day)Twice weekly |
| Spatial Resolution | 0.25° | 1km | 500m | ±300m |
| Period covered | 1997-present | 2001-2006 | 2001-present | 2006-2008 |
| Mean BA (Mha) | 346.8 | 398.9 | 360.4 | 368.3 |
| Reference | Giglio et al. (2013) | Tansey et al. (2008) | Roy et al. (2008) | Alonso-Canas and Chuvieco (2015) |

**Figures**

[Figure]

Fig. 1: Summary of the interactions between the controls on fire occurrence on coarse scales. Green filled boxes
show controls influencing fuel; blue influencing moisture; and purple influencing ignitions. Red outlined box
indicates positive influence on fire; blue a negative influence, and brown a mixed response. Brown arrows indicate
interactions between people and other controls; dark green between vegetation and other controls; and dark blue
from climate; black arrows show direct effects and red. Red arrows show feedback from fire. The arrow from
fragmentation to fuel load indicates its effect on fuel continuity.

[Figure]

Fig. 2: Summarising the levels of model complexity required to derive different aspects of global fire regimes. Outputs from models functioning at level 1 can be used to derive higher-level outputs, but it is not possible to work backwards (i.e. empirical relationships between burnt area and environmental drivers will not allow for assessment of changes in fire number and fire size). Currently there are fire routines in global DGVMs that represent all of these levels of complexity (see Table 1).), and it remains to be decided how much detail is required.

[Figure]

Fig. 3: Coefficient of variation (%) characterizing a) inter-product variability in mean burnt area; b) the inter product variability of the interannual variability in burned area; and c) the interproduct variability of the slope of temporal trends (2001-2007). Plots a) and b) are based on all four burnt area products (GFED4, MCD45, L3JRC, Fire_cciESA MERIS) whereas plot c) is based on three products and does not include the MERIS data because it is currently only available for 3 years, see Table 3.